# Landsat Data Based Prediction of Loblolly Pine Plantation Attributes in Western Gulf Region, USA

**Chongzhi Chen** [1], **Ke Wang** [1], **Luming Fang** [2], **Jason Grogan** [3], **Clinton Talmage** [3] and **Yuhui Weng** [3,*]

1. College of Environmental and Resource Sciences, Zhejiang University, Hangzhou 310058, China
2. School of Information Engineering, Zhejiang A&F University, Hangzhou 311300, China
3. Arthur Temple College of Forestry and Agriculture, Stephen F. Austin State University, Nacogdoches, TX 75965, USA
* Correspondence: wengy@sfasu.edu

**Abstract:** The suitability of using Landsat sensor variables to predict key stand attributes, including stand average dominant/codominant tree height (HT), mean diameter at breast height (DBH), the number of trees per hectare (NT), basal area per hectare (BA), and stand density index (SDI), of intensively managed loblolly pine plantations in the Western Gulf Region at the plot/stand level was assessed. In total, thirty Landsat sensor variables including six original bands, three vegetation indices, three Tasseled Cap transformed indices, and eighteen texture measure variables were used as predictors. Field data of 125 permanent plots located across east Texas and western Louisiana were used as reference data. Individual trees of those plots were measured at plot establishment (referred to as the first cycle measurement; average about 4.5 years old) and remeasured in three-year intervals (the second cycle measurement at approximately seven years old and the third cycle measurement at approximately 10 years old). Thus, field reference data represent stand development from open- (first cycle) to closed-canopy (third cycle). Models to predict stand HT, DBH, NT, BA, and SDI were developed by cycle using multiple linear regression (MLR) and also random forests (RF) methods. Results indicated that the first cycle stands HT, DBH, BA, and SDI were well predicted using the Landsat sensor variables with $R^2 > 0.7$ and low RMSEs. These relationships weakened with stand age, although still moderate with $R^2$ being around 0.45 for the second cycle measurement and became practically useless ($R^2 < 0.30$) for the third cycle measurement. For NT, no meaningful models were achieved regardless of the measurement cycle. The MLR and RF models were comparable in accuracy and had similar key predictors. Overall, the shortwave infrared bands, red band, and wetness index were the most important predictors, but their dominance declined with the cycle. Texture measure variables were relatively less important but a trend of increasing their importance with cycle was noted. Results show promise for operationally predicting stand variables for young pine plantations, an age class that typically presents significant challenges using conventional forest measurement methodologies. Potential methods to further improve model accuracy and how to use the results within the context of pine plantation management planning in the region were discussed.

**Keywords:** loblolly pine; forestry inventory; Landsat series; regression analysis; random forests

## 1. Introduction

Accurate forest inventory data are essential in forest management planning. Conventional, forest inventory data are collected via ground-based survey methods [1]. These methods, while collecting information with acceptable accuracy, have been criticized for their high cost, time consumption, and inability to update in a timely manner [1]. Indeed, exploring methods that avoid the disadvantages of ground-based methods and achieve an acceptable accuracy is of great interest to foresters. Remote sensing techniques likely are a viable alternative.

While numerous sources of remotely sensed data are available, the Landsat sensors are still the most widely used in forestry [2,3]. This is because the Landsat sensor series data, in

addition to their potential relationships with forest attributes, provide a large, free archive of imagery with medium spatial resolution, relatively large spatial extent, and frequent revisit time. Great efforts have been made to evaluate the suitability of using Landsat sensor data including individual spectral bands, vegetation indices, transformed images, and textural images, to estimate forest attributes. Mixed results have been reported, with some revealing a high level of accuracy (i.e., $R^2 > 0.60$) [4–6], and others reporting great uncertainties [7,8]. Landsat sensor variables are sensitive to many factors such as stand structure, geographic region, and human disturbance, compromising their application in predicting forest attributes. Depending on stand attribute and condition, often Landsat images were found to estimate forest stand attributes with 30% or greater error [2,9], higher than the allowable error (15~20%) of what is often required for operational forest management. Furthermore, most of the previous studies have focused on natural forests with complicated stand structures of multiple species, age classes, various densities, and understory conditions.

Little research has been dedicated to commercial plantations despite their economic importance. Compared to natural forests, the application of remotely sensed data to plantation forests potentially faces different problems [10]. Most plantations are intensively managed, single-species monocultures, resulting in distinct stand structural and temporal characteristics. Silvicultural activities such as implementing site preparation, competition control, and fertilization homogenize the understory of young (open crown) plantations, facilitating the relation between remote sensing data and stand attributes, but homogenize stand canopy structure and enhance stand productivity, compromising the ability of images in discerning canopy-closed plantations. Stand structure is a complex combination of several parameters such as stand average height, diameter, density, biomass, and volume. Previous studies have focused on using remotely sensed data to predict forest stand biomass and volume. For plantation forests, biomass and volume are secondary variables, often calculated from traits such as stand tree size (average height and diameter) and density. Therefore, the usefulness of Landsat sensor variables to recover plantation stand attributes should be evaluated by their adequacy in predicting stand tree size and density. Density management is paramount for the success of managing plantations. Previous studies on the use of Landsat sensor data have unanimously adopted the number of trees per unit area (stocking) as an indicator of density, which, however, has intrinsic limitations when applied to plantations since it does not consider tree size [11]. Basal area or stand density index combines stocking and tree size and has been widely adopted in decision-making in managing plantation density by foresters [1]. Thus, the suitability of using Landsat sensor data to predict basal area or stand density index is of great practical value, which, however, has seldom been investigated. Recent research [11] reported promising results of using Landsat sensor data to predict the height and basal area of young spruce plantations. However, much less optimistic findings were indicated for predicting the number of trees per hectare and volume of coniferous plantations [10].

Plantation management planning schedules activities aimed at optimizing the utilities of a plantation, in particular wood production. Often the key activities include, but are not limited to, projection of stand conditions and economic value (growth and yield modeling), mid-rotation silvicultural treatments, and harvest. Timely inventory data are needed to support these activities. These data should be sufficiently accurate, the level of which depends on the activity. Less precise data may be acceptable for projection purposes while a higher accuracy is desirable for supporting silvicultural activities [9]. To integrate Landsat sensor data into forest management, their usefulness must be assessed by how well they can support the activities, which, however, is missing in the literature. Most previous studies have adopted a simplistic view of the information needs in the forest planning process, without relating the analysis to management decisions [9].

Loblolly pine (*Pinus taeda* L.) is the most widely planted commercial species in the West Gulf (WG) Region. For example, in east Texas, forestland occupies about 4.9 million hectares, of which, 1.2 million hectares (24%) are classified as pine plantations, with most being

composed of loblolly pine [12]. Intensive silvicultural activities such as competition control, chemical site preparation, tillage, fertilization, thinning, and other activities have been routinely applied to loblolly pine plantations [13], resulting in rapid stand development including canopy closure over a short period post-establishment. For current operational forest management, forest inventory data are needed for young (around five years) and mid-rotation ages (around 10 years), yet traditional methods are costly. Landsat sensor data could be useful for this purpose, yet has not been fully investigated. Landsat sensor data have been demonstrated useful in managing loblolly pine plantations in ways such as mapping plantations [14] and predicting leaf area index [15–18]. Only one study [19] reported the application of Landsat sensor data to estimate loblolly pine plantation age and stocking (trees per hectare) and the results were encouraging ($R^2$ = 0.70 and 0.60, respectively). None of the above studies linked results to specific planning activities, limiting operational application. Overall, the potential of Landsat sensor data for estimating pine plantation attributes has not been fully explored, yet the interests of forest companies in using remotely sensed data for forest inventory purposes remain high (Mr. Trevor Terry, personal communication).

Initiated in 1982, the East Texas Pine Plantation Research Project (ETPPRP) has been exploring models for managing loblolly pine plantations in the region (http://etpprp.hzgzsoft.com:92/ (accessed on 8 July, 2022)). In 2004, the ETPPRP began establishing permanent plots in intensively managed loblolly pine plantations across the region, which were repeatedly measured on a three-year scale, providing temporal reference ground data. The objective of this study was to analyze the suitability of using Landsat sensor data to predict loblolly pine plantation attributes from before and after crown closure. If satellite reflectance data can be sufficiently related to measured plantation stand attributes, information from this study can be incorporated into growth and yield modeling and forest management planning.

## 2. Materials and Methods

### 2.1. Field Data

Between 2004 and 2007, permanent plots were installed in loblolly pine plantations in East Texas and into Western Louisiana to best represent the growing conditions unique to the WG region. These plantations are artificially planted, even-aged monocultures and intensively managed, and belong to a paradigm of high-yielding, short-rotation, coniferous forest plantations. Plots were generally located between 30 to 33° north latitude and 93 to 96° west longitude (Figure 1). The region belongs to the humid subtropical climate, receiving an annual average rainfall of 123 cm, paired with a mean temperature range from 2.2 °C in January to 33.9 °C in July, and an elevation ranges from 46 to 107 m, and mostly sandy acidic loamy soils.

At each study location, one square plot measuring approximately 0.1 ha (30.48 × 30.48 m) was installed [20]. A Trimble Juno SB handheld GPS receiver was used to record the latitude and longitude of the center of each plot. With ArcPad installed on the handheld, an average of 30 positions were used for capturing each point location. No post-processing differential correction was performed. It achieved a horizontal accuracy of 2–5 m HRMS. Plots were operationally managed along with the plantations, i.e., being thinned and/or fertilized when the plantations were thinned and/or fertilized. At each plot, all planted loblolly pine trees and also volunteer pine trees with a diameter at breast height larger than four inches (10.2 cm) were permanently tagged. Trees were measured when the plot was installed (referred to as the first cycle measurement) and remeasured in summer on a three-year interval (referred to as second, third, ..., cycles) for diameter at breast height (nearest 0.1 inches) and total height (nearest 1.0 foot), among other traits. Since plots were established at different years, multiple-year data were involved for each cycle (i.e., the first cycle data ranged from 2004 to 2007 and the second cycle from 2007 to 2010). Original English unit values were converted to metric values prior to analysis. Individual tree basal area was calculated using tree diameter (=0.0000785 × diameter$^2$). Individual tree data

were first examined, outliers removed, and then summarized to obtain plot and stand level mean values such as the height of the dominant and codominant trees (using the tallest 10 trees per plot) (HT in m), plot mean diameter (DBH in cm), number of trees per hectare ($NT\ ha^{-1}$) and basal area per hectare ($BA\ m^2\ ha^{-1}$). Stand density index (SDI) was calculated as $SDI = NT \times (Dq/25.4)^{-1.605}$ [21], where Dq is the stand quadratic mean diameter in cm and calculated as $Dq = \sqrt{(BA/(0.0000785 \times NT))}$. Plantation age (years) was determined as the time between the current measurement date and the plantation establishment date derived from stand records.

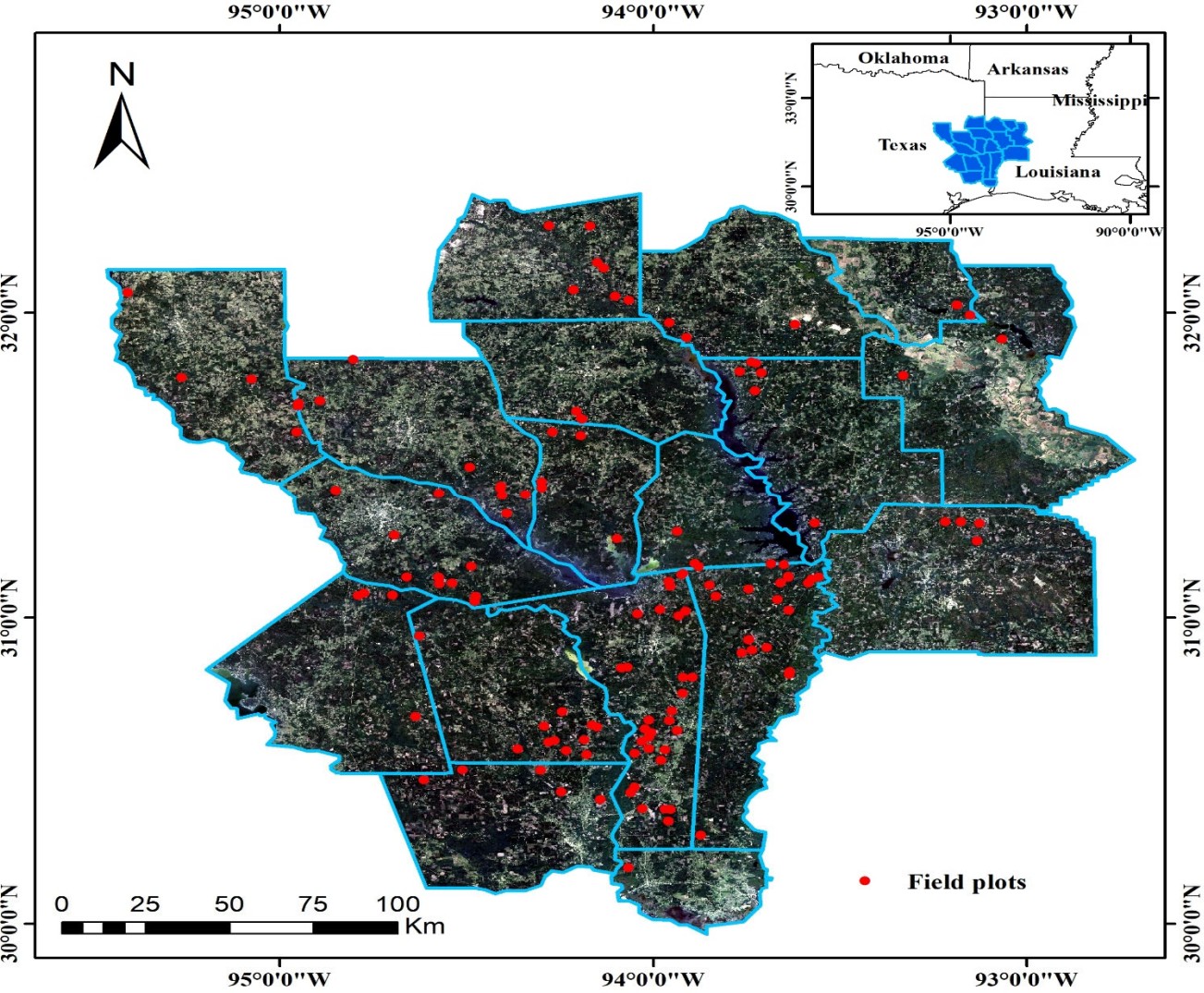

**Figure 1.** Geographic location of field permanent plots of loblolly plantations.

To date, five measurement cycles have been completed for most plots. Since this study focused on using Landsat sensor data to support two critical management activities for pine plantations (projecting stand development and guiding mid-rotation silviculture), only the first three measurement cycles were used. Data of the first cycle represent a scenario of open-canopy plantations, the second cycle of partly open-canopy plantations, and the third cycle of closed-canopy plantations. Note that plot data were further cleaned by cycle; for example, while plots of the first cycle with age = one year were excluded from the first cycle data, those plots might be included in subsequent cycles. Additionally, note that four plots have been thinned in the third cycle. Details of summary statistics of the dataset by cycle are provided in Table 1. In total, data of 116, 125, and 121 plots were used in the first, the second, and the third cycles, respectively. When the plots were established (the 1st cycle),

tree age ranged from two to 11 years old (average 4.5 years) and stocking from 829 to 2121 (average 1290) trees ha$^{-1}$. During the period of six years from age 4.5 to 9.9 years, the stand grew rapidly; the average increased from 5.4 to 9.9 m for HT, 7.1 to 15.2 cm for DBH, and 6.6 to 23.3 m$^2$ ha$^{-1}$ for BA. Additionally, during the period, the density increased substantially from 194.2 to 557.5 in SDI, and the stand canopy advanced from open to closed. While many attributes can be used to describe stand structure, the response variables in the study included those which are often required in plantation planning: HT, DBH, BA, NT, and SDI [22].

**Table 1.** Summary statistics of field reference data by cycle.

| Cycle | Attribute | Mean | Standard Deviation | N | Minimum | Maximum |
|---|---|---|---|---|---|---|
| 1 [a] | Age (years) | 4.46 | 1.79 | 116 | 2 | 11 |
| | HT (m) | 5.37 | 2.11 | 116 | 2.05 | 13.28 |
| | DBH (cm) | 7.11 | 3.34 | 116 | 2.05 | 15.07 |
| | BA (m$^2$ ha$^{-1}$) | 6.61 | 5.71 | 116 | 0.26 | 28.14 |
| | NT (trees ha$^{-1}$) | 1289.72 | 267.67 | 116 | 828.82 | 2120.49 |
| | SDI | 194.21 | 143.99 | 116 | 15.42 | 702.10 |
| 2 | Age (years) | 6.93 | 1.93 | 125 | 3 | 14 |
| | HT (m) | 8.98 | 2.15 | 125 | 3.60 | 14.97 |
| | DBH (cm) | 12.11 | 2.72 | 125 | 3.79 | 18.48 |
| | BA (m$^2$ ha$^{-1}$) | 16.08 | 6.49 | 125 | 1.52 | 32.7 |
| | NT (trees ha$^{-1}$) | 1298.38 | 255.29 | 125 | 818.06 | 2098.96 |
| | SDI | 413.28 | 144.34 | 125 | 58.34 | 790.92 |
| 3 [b] | Age (years) | 9.94 | 1.97 | 121 | 6 | 17 |
| | HT (m) | 11.67 | 2.18 | 121 | 6.67 | 21.26 |
| | DBH (cm) | 15.16 | 2.53 | 121 | 9.15 | 27.54 |
| | BA (m$^2$ ha$^{-1}$) | 23.31 | 6.10 | 121 | 8.18 | 36.10 |
| | NT (trees ha$^{-1}$) | 1263.74 | 291.31 | 121 | 365.97 | 2066.67 |
| | SDI | 557.51 | 129.71 | 121 | 225.76 | 859.39 |

[a] 7 plots with ages between 8 to 11 years; [b] 4 plots have been thinned once.

### 2.2. Satellite Data

Google Earth Engine (GEE) platform is a cloud-based platform providing a plethora of satellite images including Landsat sensor time series data from the United States Geological Survey (USGS, http://earthexplorer.usgs.gov/ (accessed on 18 June 2022)). Researchers can use GEE for pre-processing and extracting large datasets for free, which has been widely used in recent years [23].

We used longitude and latitude of a plot center to match the spatial location of the plot and its remote sensing data. We used the 30 m Collection1, Tier1, Surface Reflectance (C01/T1_SR) Landsat sensor time series data including Landsat 5 Thematic Mapper (TM), Landsat 7 Enhanced Thematic Mapper Plus (ETM+), and Landsat 8 Operational Land Imager (OLI) from GEE, depending on availability, to create satellite datasets. Thus, the sizes between reference plots and Landsat image pixels were similar to avoid spatial errors as much as possible. The C01/T1_SR datasets were geometrically and atmospherically corrected using Landsat Disturbance Adaptive Processing System (Landsat 5 and Landsat 7) [24] and the Land Surface Reflectance Code algorithm (Landsat 8). In addition, all the clouded, shadowed, and snowed pixels were masked out using QA bands with C Function of the Mask algorithm (CFMask) [25,26]. Since reference plots were installed from 2004 to 2007, Landsat sensor data were obtained correspondingly to match year of field data collection. Since field data were collected in summer, images were acquired for an early July through late August timeframe (Table 2). Most images were cloud free or had cloud cover of less than 5%. Excessively clouded or poor-quality data were disqualified and the neighboring time data instead or an alternative Landsat source were used to insure complete Landsat data coverage.

**Table 2.** Path/Row and dates of acquiring Landsat image data.

| Year | Sensor | Path/Row | | | | |
|------|--------|----------|----------|----------|----------|----------|
| | | **P24/R38** | **P24/R39** | **P25/R38** | **P25/R39** | **P26/R38** |
| 2004 | ETM+ | 9-August | 24-July | 15-July | 16-August | 7-August |
| 2005 | ETM+ | 11-July | 11-July | 19-August | 2-July | 26-August |
| 2006 | TM | 7-August | 22-July | 30-August | 30-August | 5-August |
| 2007 | ETM+ | 2-August | 2-August | 9-August | 9-August | 15-July |
| 2008 | ETM+ | 4-August | 4-August | 26-July | 26-July | 1-July |
| 2009 | TM | 15-August | 14-July | 22-August | 22-August | 12-July |
| 2010 | TM | 2-August | 2-August | 9-August | 25-August | 31-July |
| 2011 | TM | 5-August | 5-August | 28-August | 28-August | 3-August |
| 2012 | ETM+ | 30-July | 30-July | 22-August | 5-July | 28-July |
| 2013 | OLI | 9-July | 9-July | 17-August | 17-August | 8-August |

Original bands used include BLUE (450–520 nm), GREEN (520–560 nm), RED (630–690 nm), Near Infrared (NIR, 770–900 nm), Shortwave Infrared 1 (SWIR1, 1550–1750 nm), and Shortwave Infrared 2 (SWIR2, 2080–2350 nm) bands. Three commonly used vegetation indices, including Normalized Difference Vegetation Index (NDVI) [27], Enhanced Vegetation Index (EVI) [28], and Modified Soil Adjusted Vegetation Index (MSAVI) [29], were derived and used. Tasseled Cap (TC) transformation was applied for each image to produce brightness, greenness, and wetness indices [30]. Eighteen texture features were calculated by the Gray Level Co-occurrence Matrix (GLCM) (Table 3) of NDVI for each pixel with $3 \times 3$, $5 \times 5$, and $7 \times 7$ windows [31,32], respectively, but only those with $5 \times 5$ pixels were reported since adding the other two did not change results substantially. The GLCM method is one of the most commonly used approaches for textural analysis in forestry applications [33]. Thus, a total of 30 Landsat image variables were used in this study as candidate predictors (Table 3).

**Table 3.** The variables derived from the Landsat series images.

| Variables | Description | Number of Variables |
|-----------|-------------|:---:|
| Original band | Band 2 (450–520 nm): BLUE, band 3 (520–560): GREEN, band 4 (630–690): RED, band 5 (770–900): Near Infrared (NIR), band 6 (1550–1750): Shortwave Infrared 1 (SWIR1), band 7 (2080–2350): Shortwave Infrared 2 (SWIR2) | 6 |
| Vegetation indices | Normalized difference vegetation index (NDVI = (NIR − RED)/(NIR + RED)) [27]; Enhanced vegetation index (EVI = (NIR − RED)/(NIR + 6 (RED) − 7.5 (BLUE) + 1)) [28]; Modified soil adjusted vegetation index (MSAVI = NIR + 0.5 − $((\text{NIR} + 0.5)^2 - 2 \times (\text{NIR} - \text{R}))^{\frac{1}{2}}$) [29] | 3 |
| Tasseled Cap Transformation | Brightness, Greenness, Wetness [30] | 3 |
| Texture measures | Grey-level co-occurrence matrix-based texture measures including the angular second moment (ASM), contrast (CONTRAST), correlation (CORR), variance (VAR), inverse difference moment (IDM), sum average (SAVG), sum variance (SVAR), sum entropy (SENT), entropy (ENT), difference variance (DVAR), difference entropy (DENT), information measure of Corr.1 (IMCORR1), information measure of Corr.2 (IMCORR2), max Corr. Coefficient (MAXCORR), DISSimilarity (DISS), INERTIAtia (INERTIATIA), cluster shade (SHADE) and cluster PROMinence (PROM) features of NDVI for pixel size with $5 \times 5$ [31,32] | 18 |

*2.3. Model Development and Validation*

Both multiple linear regression (MLR) and random forests (RF) [34] methods were used to develop models predicting the stand attributes using the Landsat variables. These methods were selected because MLR is the most popular, while RF is the most accepted non-parametric method in forest statistical data analysis [35]. Data were split into two datasets, one for model development (75% of the data, referred to as the training dataset) and the other for model tests (25% of the data, referred to as the test dataset).

For MLR, a candidate model was first selected using the training dataset and stepwise selection, with a significant level of 0.05 being required to allow a variable into and stay in the model. To mitigate the issue of multicollinearity, the model predictors' variance inflation factors were calculated and some predictors with large values were dropped (a maximum value of five was applied). The lm function in R [36] was used for the analysis.

For RF, models were trained using the training dataset and the randomForest package in R [37]. All Landsat variables were initially included as predictors. Two algorithm parameters, the number of trees (ntree) and the number of variables per level (mtry) were tuned and results suggested a mtry of 7 and an n.trees of 500 for modeling all the attributes. The models were trained, and the importance of predictors was evaluated by calculating %IncMSE, which shows mean decrease in accuracy of the predictions when a variable is removed from the model. In the subsequent step, only those with %IncMSE $\geq$ 2 were used as predictors and used to train the final models. An %IncMSE $\geq$ 2 is an arbitrary value, which, however, simplifies the models and can increase the use by foresters. This seems reasonable since our preliminary analyses show that including predictors with %IncMSE < 2 did not change model predictability.

For the resulting MLR and RF models, the model goodness-of-fit to the training dataset was evaluated by calculating the model coefficient of determination ($R^2$) and root mean standard error (RMSE). Note that for RF models, the goodness-of-fit parameters for RF were automatically based on out-of-bag cross-validation, leading to underestimates. Models with larger $R^2$ and smaller RMSE have greater predictability, although the latter is a better choice than the former for comparing the accuracy among different models. For MLR models, residual plots were visually checked to assess model assumptions (normality, equal variance, and independence).

All final models were then tested based on the independent test dataset. $R^2$ and mean absolute error (MAE) were calculated by using model predicted and field observed values as indicators of model accuracy. MAE is preferred over RMSE in model tests due to its ease of interpretation. We did not calculate relative RMSE or MAE (express RMSE or MAE as a percentage of the sample average) since it can be misleading in repeated data since attributes with larger averages tend to have lower relative values.

## 3. Results

Selected MLR models for predicting the stand attributes using the Landsat sensor variables are shown in Table 4. The model's goodness-of-fit varied with cycle and stand attributes. For the first cycle measurement, HT, DBH, BA, and SDI were predicted well. The achieved $R^2$ ranged from 0.61 (DBH) to 0.74 (SDI) and their respective RMSE were fairly low, 1.22 m, 2.11 cm, 3.23 m$^2$, and 78.16 for HT, DBH, BA, and SDI, relative to their respective minimum (Table 1). The selected model predictors were similar for all the attributes (Table 4); the SWIR1 was consistently the top predictor for all the attributes, followed by RED (for HT, BA, and SDI) or NIR (for DBH). The lower SWIR1 or larger RED values were associated with faster stand growth and higher density. No index variables stayed in any models other than the texture measures of ASM and IDM for predicting BA and SDI. The goodness-of-fit for the HT, DBH, BA and SDI models of the second cycle measurement were greatly reduced, but still moderate, with $R^2$ ranging from 0.43 (HT) to 0.48 (SDI). SWIR1 remained the top predictor for all the models, while indices SAVG (for DBH and BA) and NDVI (for BA and SDI) were also selected. The model goodness-of-fit of the third cycle models worsened further to the point that the Landsat sensor variables were

practically useless for predicting HT and DBH. For BA and SDI, the models, respectively, explained 30% and 21% of the total variation and SWIR1 and RED were still the best predictors. Residual plots showed that model assumptions (normality, equal variance and independence) were acceptable in general, although some imperfections were present (data not shown). Differently, the NT was poorly predicted within the training datasets for all cycles, with $R^2$ being 0.19, 0.08 and 0.17 for cycles one to three, respectively. Identified predictors varied with cycle and neither SWIR1 nor RED was important predictor.

**Table 4.** Final multiple linear regression models by cycle and attribute including model predictors, model goodness of fit on the training data, and model validation on the test data.

| Cycle | Attribute | Predictors | Training | | Test | |
|---|---|---|---|---|---|---|
| | | | $R^2$ | RMSE | $R^2$ | MAE |
| 1 | HT | SWIR1, RED | 0.68 | 1.22 | 0.66 | 0.87 |
| | DBH | SWIR1, NIR | 0.61 | 2.11 | 0.61 | 1.49 |
| | BA | SWIR1, RED, ASM, IDM | 0.72 | 3.23 | 0.74 | 1.77 |
| | NT | DISS, Brightness, DVAR, NIR | 0.19 | 237.08 | 0.19 | 184.65 |
| | SDI | SWIR1, RED, ASM, IDM | 0.74 | 78.16 | 0.79 | 52.49 |
| 2 | HT | SWIR1 | 0.43 | 1.68 | 0.35 | 1.20 |
| | DBH | SWIR1, SAVG, BLUE | 0.45 | 3.43 | 0.36 | 2.57 |
| | BA | SWIR1, NDVI, SAVG | 0.46 | 4.59 | 0.52 | 3.63 |
| | NT | NIR, BLUE, Brightness | 0.08 | 236.23 | 0.10 | 182.66 |
| | SDI | SWIR1, NDVI, SHADE | 0.48 | 102.56 | 0.52 | 78.71 |
| 3 | HT | NA | NA | NA | NA | NA |
| | DBH | RED, Wetness, SAVG, Brightness | 0.08 | 2.47 | 0.26 | 1.48 |
| | BA | SWIR1, RED, NDVI | 0.30 | 5.24 | 0.14 | 4.23 |
| | NT | SWIR1, VAR, PROM | 0.17 | 253.15 | 0.23 | 180.78 |
| | SDI | SWIR1, RED | 0.21 | 117.13 | 0.33 | 81.83 |

Applying the models to the test dataset confirmed the MLR model fitness (Table 4; Figures 2–4). The obtained $R^2$ values were similar in magnitude to respective ones in the model training for each cycle. The MAE estimates were low, e.g., being 0.87 m, 1.49 cm, 1.77 m$^2$ ha$^{-1}$, and 52.49 for the first cycle HT, DBH, BA, and SDI, respectively. Overestimations occurred for small values and underpredictions were true for large values, and this trend was more evident for models of the latter cycles (Figures 2–4).

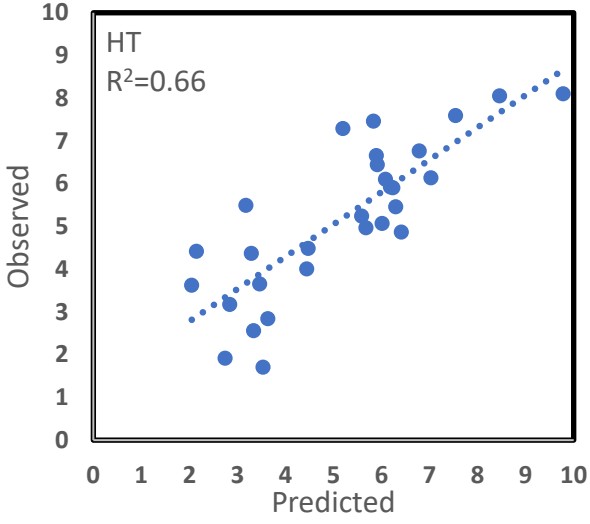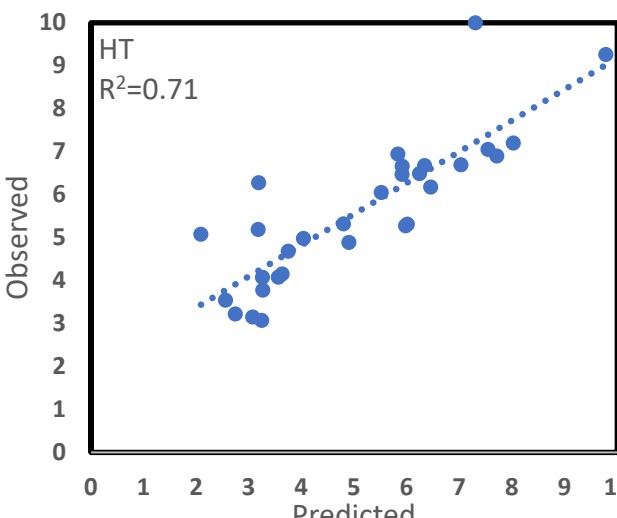

**Figure 2.** *Cont.*

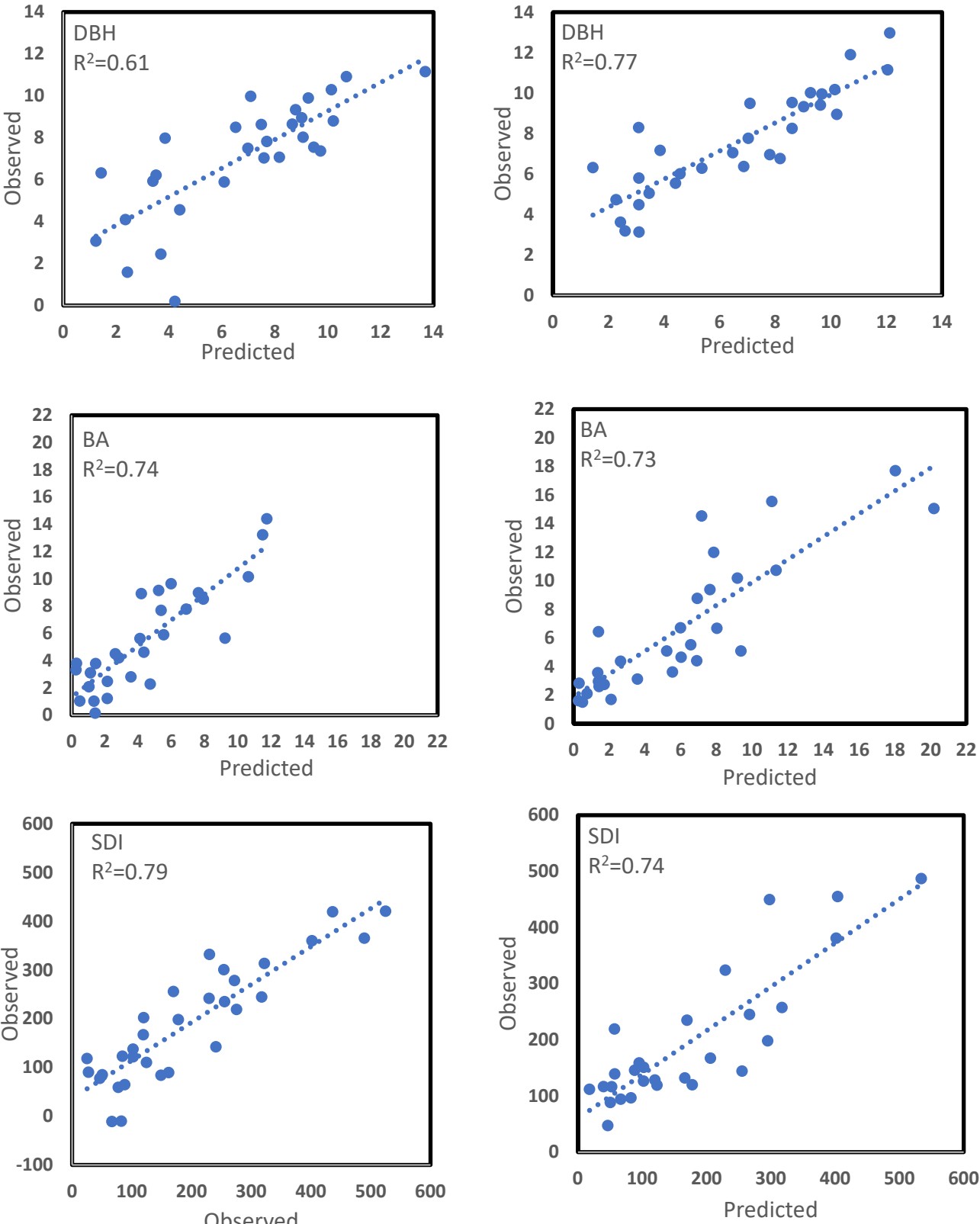

**Figure 2.** Relationships between observed and predicted values based on the independent test dataset of the 1st cycle measurement: multiple linear regression models on left column and random forests models on the right column.

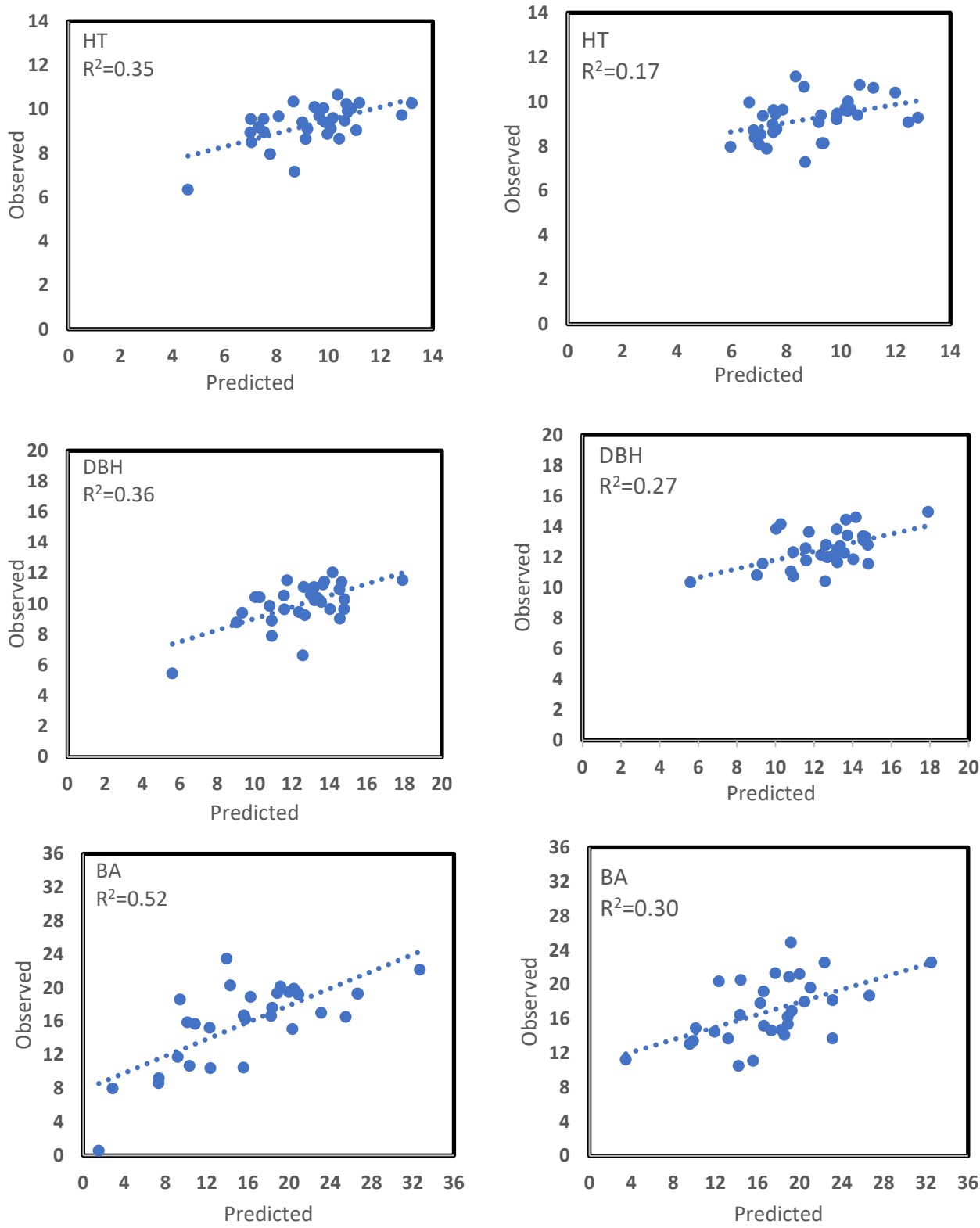

**Figure 3.** *Cont.*

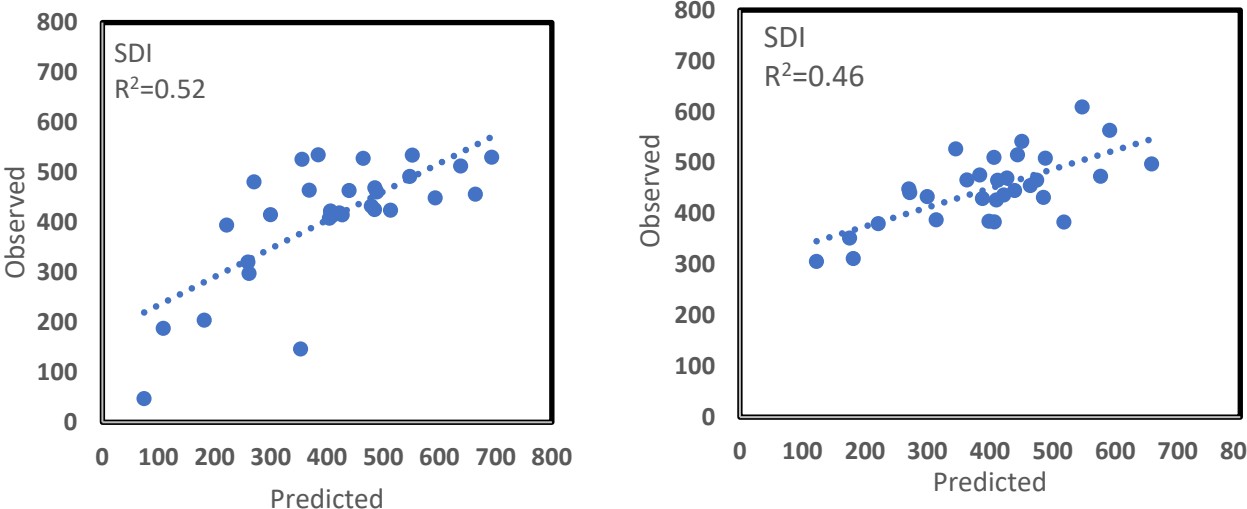

**Figure 3.** Relationships between observed and predicted values based on the independent test dataset of the 2nd cycle measurement: multiple linear regression models on left column and random forests models on the right column.

Models for predicting selected stand attributes were trained using RF to evaluate the importance of predictors, and Tables 5–7 list all the selected predictors with %IncMSE $\geq$ 2. For the first cycle measurement, the reflectance bands SWIR1, SWIR2, and RED, and the TCT variables wetness or brightness were the important (%IncMSE $\geq$ 10) predictors for modeling HT, DBH, BA, and SDI, while vegetative indices were the top predictors for predicting NT. The identified top predictors for the second cycle models remained similar, although texture index SAVG was also important for the HT, DBH, and NT models. For the third cycle, while the SWIR1, SWIR2, and wetness were still the top predictors, more texture indices were identified as the top predictors as well. Evidently, contributions were dominated by a few top predictors in the first cycle, this dominance, however, decreased with increasing cycles. For example, SWIR1 had a %IncMSE value of 30.42 for predicting BA in the first cycle, which was reduced to 24.19 and 12.53, respectively, in the second and third cycles. Many texture variables were included with %IncMSE between 2~5, only a few, e.g., SAVG and IMCORR1, had a %IncMSE > 7. The relative importance of texture variables seemly increased with the cycle. None of the vegetative indices were identified in the top four predictors for all models other than predicting NT for cycles one and two.

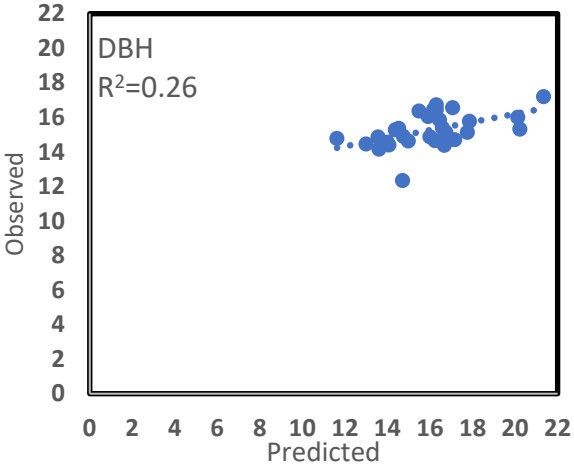

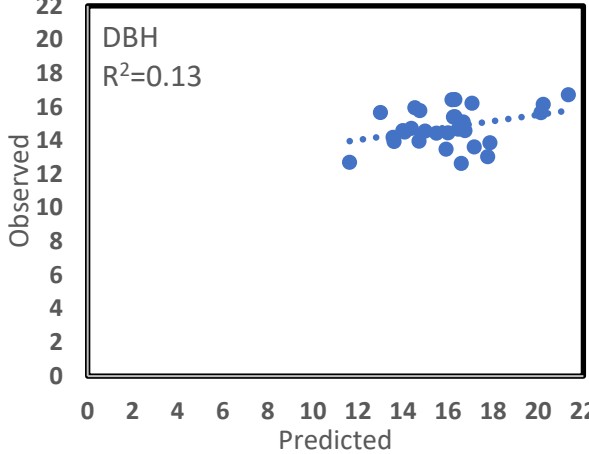

**Figure 4.** *Cont.*

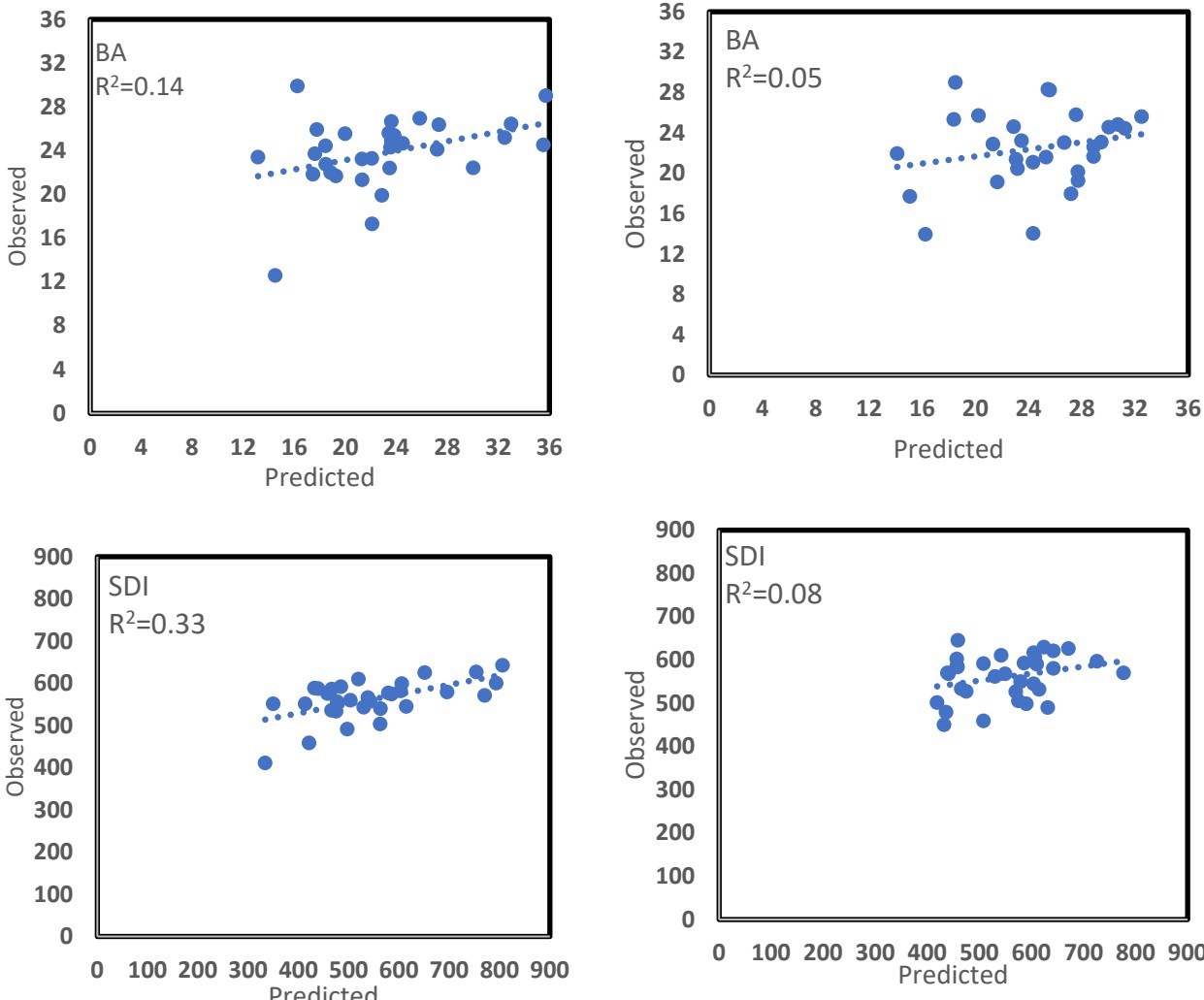

**Figure 4.** Relationships between observed and predicted values based on the independent test dataset of the 3rd cycle measurement: multiple linear regression models on left column and random forests models on the right column.

The final RF models were re-trained using the important predictors (Tables 5–7) and the training dataset, and the model goodness can be found in Table 8. For the first cycle, the models for predicting HT, DBH, BA, and SDI explained 65% or more of the total variation, and achieved reasonably low RMSEs of 1.33 m, 2.20 cm, 3.40 m$^2$ ha$^{-1}$ and 80.49 for HT, DBH, BA and SDI, respectively. The corresponding models for the second cycle showed a medium fitness with R$^2$ being around 0.40 and those for the third cycle having no practical values. Similar to the MLR results, the NT models were poor in fitness for all the cycles, with a R$^2 \leq 0.16$. Our model test using the test dataset showed that the models of the first cycle had the accuracy as expected, as demonstrated by the similar R$^2$ between training and test datasets and the low MAEs of 0.84 m, 1.34 cm, 2.06 m$^2$ ha$^{-1}$ and 60.2 for HT, DBH, BA and SDI, respectively (Table 8; Figures 2–4). However, the models of the second cycle had lower test R$^2$ than respective ones of the model fitting for all attributes other than SDI, of which a similar R$^2$ was achieved. Similar to the MLR models, the attributes were overestimated for low observed values and underestimated at the high end for all the attributes; this trend was more evident for latter cycles (Table 8; Figures 2–4).

**Table 5.** Relative importance of predictors with %IncMSE ≥ 2 by random forests models using the training dataset for the 1st cycle measurement.

| HT | | DBH | | BA | | SDI | | NT | |
|---|---|---|---|---|---|---|---|---|---|
| Variables | %IncMSE | Variables | %IncMSE | Variables | %IncMSE | Variables | %IncMSE | Variables | %IncMSE |
| SWIR1 | 28.35 | SWIR1 | 26.46 | SWIR1 | 30.42 | SWIR1 | 33.57 | MSAVI | 9.31 |
| SWIR2 | 19.37 | SWIR2 | 18.46 | SWIR2 | 23.40 | SWIR2 | 21.45 | NIR | 8.76 |
| Wetness | 11.99 | Wetness | 12.86 | RED | 11.45 | RED | 12.13 | EVI | 7.58 |
| RED | 11.01 | RED | 12.32 | Brightness | 11.36 | Wetness | 9.12 | NDVI | 6.84 |
| GREEN | 7.77 | IMCORR1 | 8.24 | GREEN | 6.73 | Brightness | 7.58 | Greenness | 6.30 |
| ASM | 6.38 | GREEN | 7.24 | NDVI | 6.29 | SAVG | 5.58 | IDM | 6.24 |
| SENT | 5.73 | IMCORR2 | 6.03 | Wetness | 5.56 | NDVI | 4.41 | ENT | 4.78 |
| IMCORR2 | 5.67 | SENT | 5.73 | ASM | 4.53 | ASM | 3.80 | Wetness | 4.37 |
| SAVG | 5.18 | ASM | 5.23 | SAVG | 4.34 | SVAR | 3.65 | PROM | 3.57 |
| Brightness | 4.83 | Brightness | 4.85 | Greenness | 4.01 | IMCORR2 | 3.56 | Brightness | 3.52 |
| Greenness | 4.67 | SAVG | 4.64 | SENT | 3.97 | GREEN | 3.47 | DISS | 3.36 |
| IMCORR1 | 4.35 | Greenness | 4.44 | IDM | 3.28 | SENT | 3.15 | SWIR1 | 3.31 |
| NDVI | 3.94 | VAR | 3.77 | IMCORR1 | 2.82 | PROM | 2.26 | IMCORR2 | 2.44 |
| DENT | 3.35 | BLUE | 2.46 | SVAR | 2.13 | ENT | 2.25 | SWIR2 | 2.38 |
| ENT | 2.86 | MSAVI | 2.12 | EVI | 2.00 | IMCORR1 | 2.10 | SHADE | 2.19 |
| BLUE | 2.48 | ENT | 2.09 | | | VAR | 2.10 | DVAR | 2.16 |
| IDM | 2.17 | | | | | | | INERTIATIA | 2.14 |
| MSAVI | 2.01 | | | | | | | | |

**Table 6.** Relative importance of predictors with %IncMSE ≥ 2 by random forests models using the training dataset for the 2nd cycle measurement.

| HT | | DBH | | BA | | SDI | | NT | |
|---|---|---|---|---|---|---|---|---|---|
| Variables | %IncMSE | Variables | %IncMSE | Variables | %IncMSE | Variables | %IncMSE | Variables | %IncMSE |
| SWIR1 | 23.97 | SWIR1 | 25.05 | SWIR1 | 24.19 | SWIR1 | 24.11 | NIR | 10.52 |
| SWIR2 | 15.20 | SWIR2 | 11.07 | SWIR2 | 17.75 | SWIR2 | 20.45 | EVI | 7.95 |
| SAVG | 7.23 | RED | 9.40 | Wetness | 7.94 | Wetness | 8.48 | SAVG | 5.30 |
| VAR | 6.58 | SAVG | 8.67 | RED | 6.14 | IMCORR1 | 8.45 | MSAVI | 4.91 |
| MSAVI | 6.19 | NIR | 6.66 | SAVG | 5.69 | IMCORR2 | 6.33 | IMCORR1 | 4.79 |
| GREEN | 5.80 | BLUE | 5.71 | IMCORR1 | 5.02 | RED | 5.78 | SVAR | 3.92 |
| NIR | 5.24 | VAR | 5.44 | PROM | 4.67 | VAR | 5.43 | SENT | 3.47 |
| Wetness | 4.42 | PROM | 4.68 | IDM | 4.03 | SENT | 5.35 | DENT | 3.42 |
| INERTIA | 3.99 | SVAR | 4.60 | IMCORR2 | 3.81 | NDVI | 3.82 | IMCORR2 | 3.19 |
| DISS | 3.89 | Wetness | 4.52 | INERTIA | 3.37 | DENT | 2.45 | DVAR | 2.75 |
| PROM | 3.76 | GREEN | 4.07 | Brightness | 3.22 | BLUE | 2.17 | DISS | 2.44 |
| DVAR | 3.73 | INERTIA | 4.02 | NDVI | 3.15 | | | INERTIA | 2.40 |
| RED | 3.48 | IMCORR2 | 3.54 | CONTRAST | 3.11 | | | CONTRAST | 2.03 |
| DENT | 3.34 | DISS | 3.49 | VAR | 2.73 | | | | |
| NDVI | 3.12 | DENT | 3.22 | DVAR | 2.71 | | | | |
| SVAR | 2.61 | NDVI | 3.06 | SVAR | 2.71 | | | | |
| CONTRAST | 2.56 | SHADE | 3.03 | SENT | 2.54 | | | | |
| EVI | 2.53 | MSAVI | 3.02 | SHADE | 2.35 | | | | |
| SHADE | 2.49 | IDM | 2.85 | DISS | 2.29 | | | | |
| BLUE | 2.19 | Brightness | 2.72 | | | | | | |
| | | CONTRAST | 2.18 | | | | | | |

**Table 7.** Relative importance of predictors with %IncMSE $\geq$ 2 by random forests models using the training dataset for the 3rd cycle measurement.

| HT | | DBH | | BA | | SDI | | NT | |
|---|---|---|---|---|---|---|---|---|---|
| **Variables** | **%IncMSE** | **Variables** | **%IncMSE** | **Variables** | **%IncMSE** | **Variables** | **%IncMSE** | **Variables** | **%IncMSE** |
| RED | 6.18 | DISS | 8.58 | SWIR1 | 12.53 | SWIR1 | 16.91 | SWIR1 | 12.02 |
| DVAR | 5.64 | SWIR2 | 6.09 | Wetness | 9.60 | Wetness | 14.43 | IDM | 8.96 |
| DISS | 5.42 | CONTRAST | 5.40 | SWIR2 | 6.36 | ENT | 6.95 | SWIR2 | 7.13 |
| SWIR1 | 5.22 | SWIR1 | 5.37 | RED | 5.96 | SWIR2 | 5.96 | ENT | 5.83 |
| INERTIA | 4.89 | INERTIA | 4.98 | NDVI | 5.64 | IMCORR2 | 5.74 | ASM | 4.84 |
| CONTRAST | 4.80 | IMCORR2 | 4.61 | ENT | 5.01 | NDVI | 4.17 | IMCORR1 | 4.64 |
| Wetness | 4.18 | ASM | 4.53 | BLUE | 4.41 | ASM | 3.88 | MSAVI | 4.38 |
| SWIR2 | 3.94 | DVAR | 4.13 | DISS | 4.08 | RED | 3.81 | EVI | 4.05 |
| IDM | 3.88 | NDVI | 4.08 | GREEN | 3.62 | IDM | 3.71 | NDVI | 3.77 |
| GREEN | 3.53 | DENT | 4.06 | IMCORR2 | 3.39 | EVI | 3.53 | Wetness | 3.64 |
| IMCORR2 | 3.43 | RED | 4.04 | DENT | 2.63 | CORR | 3.36 | RED | 3.32 |
| Greenness | 3.32 | IMCORR1 | 3.88 | INERTIA | 2.35 | IMCORR1 | 3.09 | NIR | 3.16 |
| VAR | 2.85 | SENT | 3.88 | IDM | 2.29 | DVAR | 2.50 | SENT | 3.11 |
| EVI | 2.67 | MSAVI | 3.87 | SVAR | 2.17 | DISS | 2.25 | VAR | 3.09 |
| PROM | 2.64 | ENT | 3.67 | ASM | 2.06 | VAR | 1.97 | DENT | 3.05 |
| NDVI | 2.38 | SHADE | 3.65 | | | | | SVAR | 2.54 |
| BLUE | 2.34 | GREEN | 3.49 | | | | | PROM | 2.40 |
| MSAVI | 2.10 | IDM | 3.44 | | | | | IMCORR2 | 2.02 |
| IMCORR1 | 2.08 | EVI | 3.03 | | | | | | |
| | | CORR | 2.93 | | | | | | |
| | | Wetness | 2.59 | | | | | | |
| | | VAR | 2.39 | | | | | | |
| | | SVAR | 2.36 | | | | | | |

**Table 8.** Final random forests models by cycle and stand attribute including model goodness-of-fit on the training data and model validation on the test data.

| Cycle | Attribute | Train | | Test | |
|---|---|---|---|---|---|
| | | **$R^2$** | **RMSE** | **$R^2$** | **MAE** |
| | HT | 0.68 | 1.33 | 0.71 | 0.84 |
| | DBH | 0.65 | 2.20 | 0.77 | 1.34 |
| 1 | BA | 0.69 | 3.40 | 0.73 | 2.06 |
| | NT | 0.16 | 231.11 | 0.25 | 192.75 |
| | SDI | 0.70 | 80.49 | 0.72 | 60.20 |
| | HT | 0.41 | 1.74 | 0.16 | 1.36 |
| | DBH | 0.42 | 2.22 | 0.27 | 1.38 |
| 2 | BA | 0.36 | 5.36 | 0.29 | 3.89 |
| | NT | 0.08 | 253.11 | 0.06 | 154.07 |
| | SDI | 0.42 | 114.16 | 0.46 | 85.16 |
| | HT | 0.06 | 2.46 | 0.08 | 1.61 |
| | DBH | 0.13 | 2.78 | 0.13 | 1.85 |
| 3 | BA | 0.07 | 6.25 | 0.05 | 4.98 |
| | NT | 0.08 | 288.81 | 0.09 | 222.83 |
| | SDI | 0.25 | 110.08 | 0.10 | 72.29 |

## 4. Discussion

Intensively managed loblolly pine plantations are often subjected to a variety of silvicultural treatments at the time of planting or shortly thereafter [38], and their canopy structures develop in a rapid, yet predictable, way. Intensive management enhances tree growth and suppresses underground vegetation, creating a favorable environment for using images for open-canopy plantations, but can lead to homogenous, dense stand canopy structures, weakening penetrability of Landsat sensor signals and compromising

the estimation of stand attributes after clown closure. The repeated (cycles one to three) data used represent crown development from pre- to post-closure over a period of six years in operational loblolly pine plantations. Results reveal that HT, DBH, BA, and SDI of loblolly pine plantations may be predicted adequately when crowns were open, but such predictability weakened as stands developed, being moderate when crowns were partially closed and becoming practically useless after crowns were completely closed. To our knowledge, this study is the first to quantify changes in Landsat sensor data suitability within a critical period of stand development from open crown to complete crown closure. The general belief that strong relationships exist between Landsat sensor data and stand attributes in homogeneous forests such as in conifer plantations [39,40] is an oversimplification. Since the plantation canopy was at such a high level of density after crown closure that Landsat sensors lost the ability to penetrate the canopy and were unable to differentiate between individual tree canopies, compromising the estimates of DBH, and consequently the estimates of BA and SDI. In upland conifer plantations of various ages in Scotland, stand HT and BA were strongly ($R^2 > 0.77$) correlated with Landsat sensor bands but the strength of the relationships decreased substantially after canopy closure occurred [39]. The strong predictability of Landsat sensor variables for young (pre-crown closure, 2–17 years old) Sitka spruce plantations has been reported [11]. In even-aged *Eucalyptus* plantations, correlations between high spectral resolution ASTER satellite data and stand trees ha$^{-1}$, DBH, HT, BA, and volume were stronger for plantations of ages 4–6 years than those of 7–9 years [41]. These studies paired with our findings suggest that the suitability of using Landsat sensor data to estimate plantation stand attributes varies with the stand development stage, being more useful for young and open-canopy plantations.

One important finding was that Landsat images were not practically useful for predicting NT during the study period (Tables 4 and 8), suggesting that the counting of trees is difficult when the satellite images are of moderate resolution, especially when tree canopies overlap. This disagrees with findings from a study also on loblolly pine plantations in east Texas [19] where as high as 60% of the variation in NT was accounted for by Landsat sensor variables such as NDVI. This inconsistency could partly be due to sampling; Their NT ranged from 222 to 2396 trees ha$^{-1}$, whereas our study sampled a much narrower NT range (Table 1), increasing the difficulty of discerning populations from Landsat sensor variables. Additionally, reflection of satellite on a pine plantation is more determined by crown characteristics, in particular crown width, a strong predictor for tree DBH. Thus, a stronger relationship of Landsat sensor data with BA or SDI over NT is not unexpected. Poor predictability for NT using Landsat sensor variables has been confirmed in young commercial plantations of other species [11,39]. Thus, there is uncertainty associated with the application of using Landsat variables to predict NT for commercial plantations, suggesting further investigation of other variables or methods to explain forest variability in NT is necessary. Further discussion focuses more on stand attributes of HT, DBH, BA and SDI.

While used widely, the MLR method is often criticized for model development to estimate stand attributes since several assumptions (linear relationship, equal variance, normality and independence) are required. RF has theoretical advantages over MLR modeling by overcoming the assumption-related limitations and also its ability to address the issues of high-dimensionality and multicollinearity. RF, relatively new in forest statistical data analysis, has recently been adopted to predict forest stand attributes using climate and biophysical data [35,42,43]. For this data, even with many strongly correlated Landsat variables being involved, the models from RF and MLR were competitive with one another in predictability; neither approach produced substantially better models (Tables 4 and 8). The selected predictors by both methods were similar, in particular for the first cycle models. In a study to modeling stand BA, volume and biomass using airborne Lidar and multispectral imagery in heterogeneous mixed species forests in Alabama, models developed by MLR and RF were comparable [44]. Fewer Landsat variables were selected as predictors of

the MLR models, and their predictability may be improved by including highly related predictors such as SWIR2, which, however, complicates biological explanations. Further improvements in predictability may be realized by adding extra predictors to RF models too, but the improvement is unlikely to be substantial. One drawback of using RF technique to develop models is the implementation of the models which is difficult without programming the models into source codes [35]. Currently the MLR models proposed in this study are recommended over RF models due to their ease of use, comparable accuracy and straight-forward interpretation.

Both the MLR and RF models tended to bias for low (overestimate) and high (underestimate) observed values (Figures 2–4), which has been identified in previous studies [45]. For plantations with closed crowns and high productivity, the limited sensitivity of the optical sensors for interpreting the canopy structure leads to saturated measurements while for young stands with open crowns the confounding effect of understory vegetation may overestimate stand attributes. Our results further indicated that the biased tendencies varied with stand development, being relatively minor for the first cycle measurement and becoming more evident with stand crown closing.

Landsat sensor data are commonly used for assessing stand attributes due to their potential correlations, free availability, and broad spatial and temporal coverage. While numerous Landsat variables are available, the selection of the ideal ones is challenging, varying with stand structure and environment. The dynamic changes in stand density (crown closure) and understory background as stands develop can lead to various variables being the optimal predictors at different stages. In this regard, some patterns stood out based on our results for a short period of six years of the plantation's early growth. First, the bands, SWIR and RED, were consistently the most important predictors. Previous studies on loblolly pine plantations indirectly support our findings, where SWIR1, RED, and NDMI (an index strongly influenced by SWIR) strongly correlated with stand age and leaf area index, both growth-related stand attributes [15,18,19]. The SWIR1 was also identified as the top predictor for forest stand properties on other pine stands [39,46,47]. Second, wetness could be a key variable for predicting tree size of open-canopy plantations (Table 5) and for predicting stand density of plantations with partly or completely closed crowns (Tables 6 and 7), which is in parallel to the findings on other species [19]. Our MLR models did not include wetness due to its strong correlation with SWIR, leading to an issue of multicollinearity. Third, all vegetative indices except NDVI were relatively unimportant. NDVI was selected as a predictor for predicting BA and SDI for canopy-closed plantations (Table 4). Vegetation indices are useful in estimating stand attributes such as biomass because they enhance green vegetation signals and minimize the impacts from the surface and atmospheric effects in complex vegetation structures [4]; their usefulness likely reduces in relatively simple and homogeneous structures such as intensively managed pine plantations. Fourth, many texture variables contributed somewhat to improving model prediction, with most having a %IncRME between 2–4. The models centralized the importance at a few band variables such as SWIR1 and RED before crown closure, this dominance declined as stands develop, and models split the importance to more texture measure variables after crowns were completely closed (Tables 5–7). Texture features performed better than spectral factors for modeling stand attributes in areas with multiple layers and complex forest stand structures [48,49], and our results support the pattern that the importance of texture variables increased from the first cycle in a situation of simple structure to the more complex structure of the third cycle.

Basal area (BA) and SDI are numerical values that capture intensity of competition within forest stands. Operationally, BA and SDI are parameters often used to help foresters with thinning decisions, and thus accurately predicting BA or SDI in particular at the mid-rotation age has great practical value. Even models predicting BA and particularly SDI had larger $R^2$ than the respective ones for predicting HT and DBH (Tables 4 and 8), their accuracies in predicting the mid-rotation age (third cycle) BA and SDI were low, with

$R^2 < 0.25$, suggesting infeasible of the method to guide thinning of loblolly pine plantations unless the models can be otherwise improved in future.

Can the findings of this study be incorporated into plantation management of loblolly pine in the region? This question has to be answered within the context of operational pine plantation planning. For loblolly pine plantations in the region, often a timber cruise around 5 years is carried out to obtain stand information such as mean HT, DBH, and NT, which is then entered into regional growth and yield models [22] to project future stand characteristics (BA, volume, and NT) and economic values under scenarios of various silvicultural activities. For this purpose, the required accuracies for predicting HT, DBH, and NT are relatively low. Our results are encouraging for predicting HT and DBH of the plantations around age five (Tables 4 and 8). The issue was NT, which was poorly predicted for all cycles. In reality, however, loblolly pine plantation survival is routinely surveyed at the end of the first or second growing season, and this number is unlikely to change substantially in the subsequent years until competition-induced mortality occurs [38]. In other words, NT of young loblolly pine plantations are typically known. Thus, our results do support the use of Landsat sensor data to facilitate the long-term management planning. Forest inventory data are indispensable for mid-rotation (around age 10 years) loblolly pine plantations to support silvicultural activities such as commercial thinning and/or fertilization. Toward this end, Landsat sensor data were found not to be reliable indicators of stand attributes and, therefore, caution should be taken for using Landsat sensor data to replace field measurements.

Are there options to improve the predictability of the models? With the inclusion of other factors, a few studies have shown positive results in predicting stand attributes. Ages are known accurately in any well-managed loblolly pine plantations and adding age as a predictor was reported to improve the performance of Landsat sensor data-based models [45]. In this study, when stand age was added as a predictor to the MLR models for predicting HT, DBH, BA, and SDI, the model $R^2$ was improved to 0.82, 0.75, 0.80, and 0.80 for the first cycle, 0.72, 0.62, 0.62, and 0.61 for the second cycle, and 0.55, 0.44, 0.41, and 0.32 for the third cycle (note that multicollinearity did not become an issue), greatly reducing the uncertainties of model estimates. Geographic factors, such as stand elevation, slope, and aspect, are known to affect spectral reflectance values [50], and thus including them may improve model performance. We are currently collecting this information so that a test on the topic can be carried out. Synergistic use of data from different sources or from different sampling dates (or seasons) of one source to estimate stand attributes can be more accurate than data from a single source or a single date [3,51]. For this study, Landsat sensor data were extracted for summer only. The suitability of incorporating Landsat sensor data from other seasons to improve model performance should be further investigated. Clearly, applying high-resolution remotely sensed data would improve model performance, which, however, is accompanied by higher data costs. Reference [52] estimated stand attributes of pine plantations using a WorldView-2 multispectral imager (2m spatial resolution) and achieved acceptable estimates for HT and DBH (errors of <13%) but relatively large errors (up to 30%) for BA, volume, and NT estimates. In another study, researchers estimated BA for a plantation of loblolly pine using airborne LiDAR data, achieving an accuracy of $R^2 = 0.97$ [40].

Field reference data well representing the truth are indispensable for model development. There are a few issues about field data which need to be clarified for this study. Four plots were installed in plantations with ages between 8~11 years, and therefore their first cycle data do not represent open-canopy plantations. Additionally, four plots at the third measurement cycle have been commercially thinned once. We did not exclude these plots in analyses since our preliminary analyses showed that removing them would not change the results. Our field plots were in the square of 900 m$^2$, very similar to the pixel shape and size, nonetheless, a potential mismatch between the field coordinates and the points of spectral variable extraction might occur. Note that the plots measured in this study are considered operational for loblolly pine plantations in the WG region. Consequently, our

results well represent the relationships between Landsat sensor data and observed growth and density in operational settings there, and thus, the models may be valuable tools for managing the current regional loblolly pine plantations.

## 5. Conclusions

In current industrial pine plantation management in Southern US, inventory data at the young stand development (i.e., for projecting stand development) and mid-rotation stage (for guiding silvicultural activities) are indispensable. Using repeated data collected from loblolly pine plantations in the WG region, USA, the suitability of using Landsat sensor data to predict key plantation attributes for the early (open-crown) to mid-rotation (closed crown) stage was evaluated. Landsat sensor data predicted the young, open-canopy plantation tree height, DBH, basal area, and stand density index adequately. This predictability, however, weakened with stand development, becoming moderate for plantations with partly closed crowns and poor for plantations with completely closed crowns. Regardless of stage, Landsat sensor data predicted the number of trees poorly, which, however, often has been known. The shortwave infrared bands, red band, and wetness index were the most important predictors, but their dominance declined with stand development. Texture measure variables were relatively less important, but a trend of increasing importance was noted as stands developed during the study period. Model predictability may be improved by adding other predictors such as stand age. We concluded that Landsat sensor data may be used to obtain stand-level information for projecting stand development but may not be accurate enough to guide mid-rotation silviculture activities such as thinning.

Developed models are based on the training dataset, extrapolation outside the range of the training dataset can be biased, and caution should be used. We suggest that the procedure adopted in this study should be tested in loblolly pine and other commercial pine plantations of various stages across Southern US while investigating the accuracy of using Landsat sensor series and other remote sensing data.

**Author Contributions:** Conceptualization, C.C. and Y.W.; methodology, C.C., Y.W., K.W. and L.F.; software, C.C., K.W. and L.F.; formal analysis, C.C., Y.W. and C.T.; investigation, C.C. and J.G.; resources, Y.W. and KW; writing—original draft preparation, C.C. and Y.W.; writing—review and editing, J.G., K.W., L.F. and C.T.; supervision, K.W. and Y.W.; project administration, Y.W.; funding acquisition, Y.W. All authors have read and agreed to the published version of the manuscript.

**Funding:** This research was funded by the ETPPRP and the McIntire—Stennis program.

**Acknowledgments:** We would like to thank the ETPPRP members (Rayonier, Resource Management Services, and Forest Resources Consultants) and the Stephen F. Austin State University for their support of the program as well as all the ETPPRP student workers who helped collect data.

**Conflicts of Interest:** The authors declare no conflict of interest.

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
