# Peer review of "Landsat Data Based Prediction of Loblolly Pine Plantation Attributes in Western Gulf Region, USA"

_remotesensing, doi:10.3390/rs14194702_

Round 1
Reviewer 1 Report
The manuscript is well written and presented and I think it is suitable for publishing in Remote Sensing. I have only a few minor comments.
- I am concern about the length of some sentences and paragraphs. For example, I have to read the sentence in the abstract (lines 15 to 19) two times to get the meaning. It will be much better to keep sentences as short as possible. Also, in the discussion part paragraphs are too long (first paragraph).
- Line 38. The second sentence in the introduction “Conventional, forest inventory data are collected via ground-based survey methods.” You need to insert a reference or references as an example.
- Lines 39-40. You need to refer to the reference or references who proved that the survey methods are time consuming and expansive.
- Line 157. Five measurement cycles.
- Tables and figures should be inserted close to the first citation in the text. Table 1 should be inserted before subsection 2.2 Satellite data. Similarly, table 2 should be inserted before subsection 2.3 Model development and validation. Also, tables should be presented in one page.
- Line 226. %IncMSE ≥2 better that %IncMSE>=2.
- Goodness-of-fit sometimes is written as goodness of fit without dashes. Why?
- Line 250. Must be table 3 not 1.
- Table 3 inserted inside the paragraph.
- Charts in figure 3 and 4 should be in the same size. It is better for the presentation.
Reviewer 2 Report
Dear Sirs,
I found your paper of moderate content significance, anyway useful from the methodological point of view for other researchers to avoid setting errors and to focalize the topic. You can see a few minor criticisms in attached file. I recommend pubblication after a minor revision.
Best regards

Reviewer 3 Report
The objective of this study was to analyze the suitability of using Landsat data to predict loblolly pine plantation attributes from before and after crown closure. If satellite reflectance data can be sufficiently related to measured plantation stand attributes, information of this study can be incorporated into growth and yield modeling and forest management planning. The subject of the paper is important for studying forest plantations, in this case, the loblolly pine. It will contribute to increase the literature of forest plantations. It needs more details of loblolly pine characteristics especially its growth cycle. It also needs more details about the dates of Landsat images used in this work. These information would be helpful for the readers to understand the work. What is the criteria to choose the 3 cycles ? The results would probably be better if consider only one cycle.
Some specific comments
L.87- (growth and & yield), --- (growth and yield),
Figure 1- I suggest to include the Landsat image in the figure
L.159 through the manuscript- of open-canopied plantations, the 2nd cycle of partly open-canopied plantations,, --- of open-canopy plantations, the 2nd cycle of partly open-canopy plantations,
Round 2
Reviewer 3 Report
The objective of this study was to analyze the suitability of using Landsat data to predict loblolly pine plantation attributes from before and after crown closure. If satellite reflectance data can be sufficiently related to measured plantation stand attributes, information of this study can be incorporated into growth and yield modeling and forest management planning. The subject of the paper is important for studying forest plantations, in this case, the loblolly pine. It will contribute to increase the literature of forest plantations. The revised manuscript has been improved by the authors following the reviewers comments.
Page 6 an 197 alternatiuve Landsat --- an 197 alternative Landsat
Page 20 - other remote sense data --- other remote sensing data
Author Response
Reviewer 3
The objective of this study was to analyze the suitability of using Landsat data to predict loblolly pine plantation attributes from before and after crown closure. If satellite reflectance data can be sufficiently related to measured plantation stand attributes, information of this study can be incorporated into growth and yield modeling and forest management planning. The subject of the paper is important for studying forest plantations, in this case, the loblolly pine. It will contribute to increase the literature of forest plantations. The revised manuscript has been improved by the authors following the reviewers comments.
Page 6 an 197 alternatiuve Landsat --- an 197 alternative Landsat
Corrected. See Line 198.
Page 20 - other remote sense data --- other remote sensing data
Corrected. See Line 581.